# Predicting Daily Suspended Sediment Load Using Machine Learning and NARX Hydro-Climatic Inputs in Semi-Arid Environment

Mohamed Abdellah Ezzaouini [1,2,*] , Gil Mahé [3] , Ilias Kacimi [1], Ali El Bilali [2] , Abdelaziz Zerouali [2] and Ayoub Nafii [2]

1   Geoscience, Water and Environment Laboratory, Faculty of Sciences, Mohammed V University, Avenue Ibn Batouta, Rabat 10100, Morocco; iliaskacimi@yahoo.fr
2   River Basin Agency of Bouregreg and Chaouia, Benslimane 13000, Morocco; ali1gpee@gmail.com (A.E.B.); azlso@yahoo.fr (A.Z.); ayoubnafii@gmail.com (A.N.)
3   Hydroscience Montpellier (HSM), Université Montpellier, CNRS, IRD, IMT, 34398 Montpellier, France; gil.mahe@ird.fr
*   Correspondence: ezzaouini70@yahoo.fr; Tel.: +212-661-110-897

**Abstract:** Sediment transport in basins disturbs the ecological systems of the water bodies and leads to reservoir siltation. Its evaluation is crucial for managing water resources. The practical application of the process-based model can confront some limitations noticed in the lower accuracy during the validation process due to the lack of reliable physical datasets. In this study, we attempt to apply machine-learning-based modeling (ML) to predict the suspended sediment load, using hydro-climatic data as input variables in the semi-arid Bouregreg basin, Morocco. To that end, data for the years 2016 to 2020 were used for the training process, and the validation was performed with 2021 data. The results showed that most ML models have good accuracy, with a Nash–Schiff efficiency (NSE) ranging from 0.47 to 0.80 during the validation phase, which indicates satisfactory performances in predicting the SSL. Furthermore, the models were ranked against their generalization ability (GA), which revealed that the developed models are good to excellent in terms of GA. Overall, the present study provides new insight into predicting the SSL in a semi-arid environment, such as the Bouregreg basin.

**Keywords:** suspended sediment load; generalization ability; uncertainty; Bouregreg; Morocco

## 1. Introduction

Sediment transport in hydrodynamic systems is still not a well-understood phenomenon and, therefore, it is a crucial topic in hydrological studies. Its exceedance existence in streams can significantly impact the river flows by changing the rheological proprieties of the water, which can impact the design capacity of the channel culverts and increase the damages associated with flash floods [1–4]. The siltation of sediments in dam reservoirs also reduces their storage capacity, leading to freshwater scarcity, especially in arid and semi-arid regions [5]. Moreover, sediments can pose a threat to the water bodies' quality and lead to ecological damages by disturbing aquatic systems [6,7]. Consequently, the understanding of the sediment transport mechanism and its quantification at the basin scale are challenging problems in water resources planning and management, particularly in dynamic basins under high climatic variability.

For decades, many soil erosion models have been developed and evaluated to assess the soil erosion phenomenon, such as the most known empirical-based models: the universal soil loss equation (USLE) [8]; its derivatives, modified USLE, named MUSLE [9]; and the revised one, which is called RUSLE [10]. Indeed, these models embedded in geographic information system (GIS) tools are considered as an outstanding approach to assess water erosion at the basin scale [11–13]. However, the applications of the process-based models

require several physical and hydrological parameters and characteristics of basins, which can be a source of uncertainties, especially in poorly monitored areas. Additionally, the highly dynamic basin characteristics leads to the questionable ability of the process-based model to evaluate the sediment load over time.

In poorly monitored areas, the machine learning (ML) models can overcome the shortage and variability of the physical parameters using available archived datasets. Indeed, their construction relies on the statistical input–output equation rather than explaining the mechanism involved in the process. Recently, several studies showed that the data-based models are powerful tools to overcome some limitations of the conceptual-based models in predicting the water resources status [14–18]. For instance, ref. [19] demonstrated the accuracy of the support vector machine method to estimate the shear stress in the rectangular channel. As for the prediction of the sediment transport, ref. [20] applied ensemble genetic programming to estimate the incipient sediment motions in rectangular channels, and demonstrated the superiority of the applied approach with a coefficient of correlation r of approximately 0.92. Similarly, in hydrologic studies, ref. [21] applied a long short-term memory neural network (LSMNN) to predict the suspended sediment concentration (SSC) in the Johor River in Malaysia and found a high accuracy prediction in these models. In [22], three artificial intelligence models were used to estimate the sediment load in Ethiopia. In [23], a multiple linear regression model was applied to predict the suspended sediment yield (SSY) in the Cuyahoga River in Ohio through satellite images and rainfall datasets. Importantly, all of these studies demonstrated that the ML-based models presented a high accuracy for predicting the sediment load at various basin scales.

In Morocco, water resource planning and management processes are facing several issues, such as reservoir sedimentation, the continuous decline of groundwater level in most aquifers, seawater intrusion into the costal aquifers, inappropriate practices applied to groundwater-based agriculture, and flood risk [24–26]. Importantly, according to the Water Department of Morocco, reservoir silting causes a global decrease in the reservoir capacity of approximately 70 $mm^3 \cdot yr^{-1}$,which means 0.4% each year, with large regional variability, and much higher values in some regions, such as the north of the country. Indeed, this loss is generally assessed through topo-bathymetric surveys at the reservoir's scale. Despite being an efficient approach to directly evaluate the sediment in reservoirs, this method does not allow for making decisions to reduce the soil erosion at the scale of basins. Meanwhile, the evaluation of the suspended sediment load (SSL) at basin scales could be valuable in classifying basin vulnerabilities to the soil erosion and, therefore, in prioritizing mitigation measures. The models of USLE and its derived versions are commonly applied approaches in Northern African countries to evaluate the soil erosion, including Morocco [27]. Recently, in our study [28], the comparison of two years of the sediment measurement with MUSLE-based simulation results demonstrated a low prediction accuracy, with a Nash–Sutcliffe efficiency index (NSE) lower than 0.5 [29]. Consequently, it is highly needed to test new procedures, such as ML-based modeling, to predict the daily suspended sediment load (SSL) and to improve the prediction accuracy using hydro-climatic input variables.

In this study, we apply a ML model to overcome the drawbacks of the process-based models and to improve the prediction accuracy of the daily SSL at four runoff gauging stations in the Bouregreg semi-arid basin, Morocco. This goal can be achieved by (1) developing ML models with NARX hydro-climatic input variables, (2) evaluating the prediction accuracy of the models, and (3) ranking the models for simulation purposes based on the generalization ability (GA) metric and uncertainty analysis.

## 2. Materials and Methods

### 2.1. Study Area Description

This work focuses on the Bouregreg basin, which is located in the Rabat-Sale-Kenitra and Beni Mellal-Khenifreg provinces in the Mediterranean area (Figure 1). This basin covers 9970 $km^2$ and consists of 4 basins; three main rivers form the hydrographic network,

namely: Bouregreg River (264 km long), Grou River (249 km long), and Mechraa River (132 km long). It is monitored by 9 rainfall runoff stations, as presented in Figure 1.

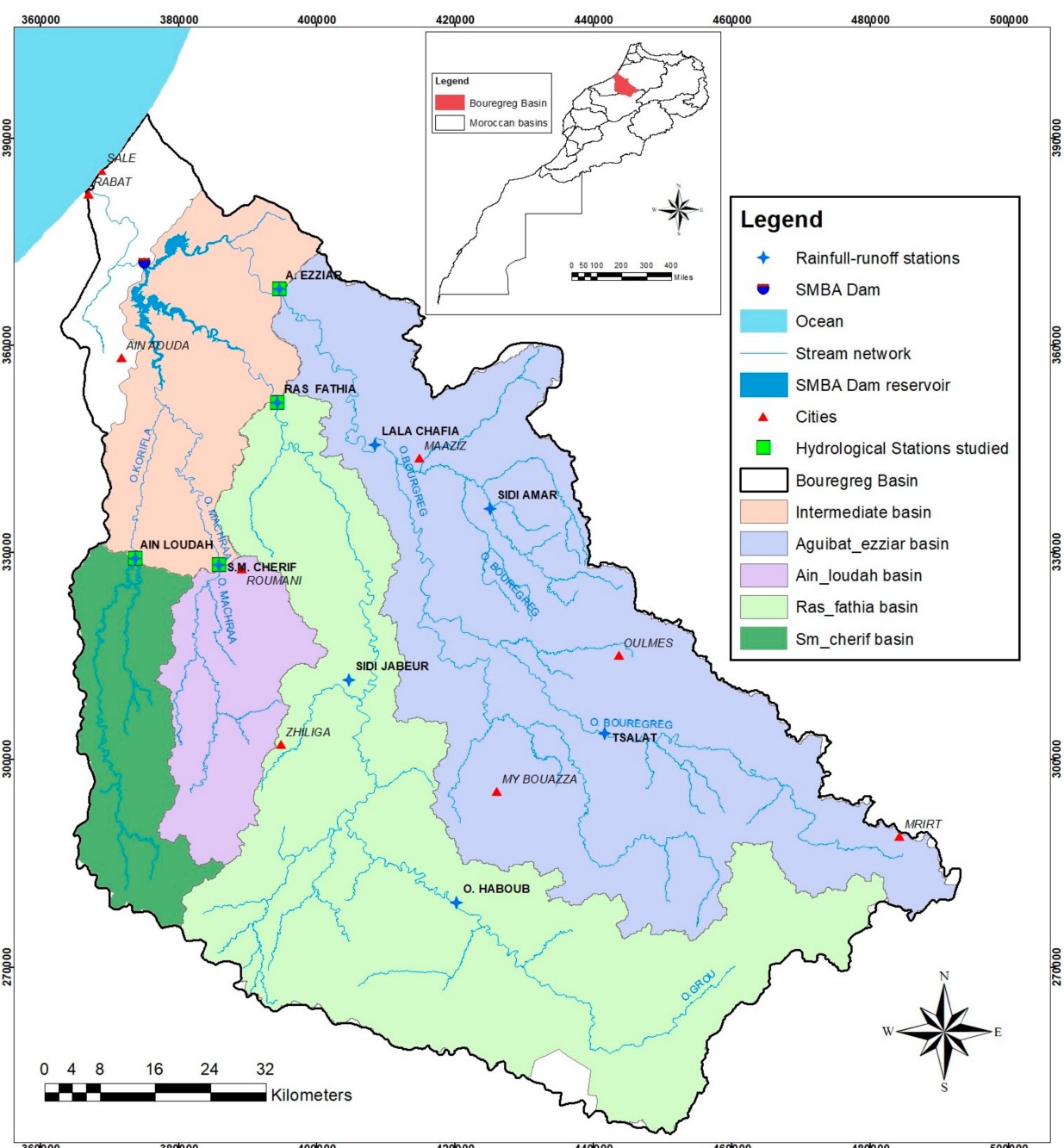

**Figure 1.** Study area location in Morocco.

However, according to the River Basin Agency of Bouregreg and Chaouia, the annual precipitation in the basin is 400 mm·yr$^{-1}$ in the northwest part and 760 mm·yr$^{-1}$ in the mountainous part, whereas the mean inflow volume of SMBA's dam reservoir is approximately 680 mm$^3$·yr$^{-1}$ (1975–2021). This reservoir has a normal capacity of 975 mm$^3$ and supplies drinking water to the coastal area between Rabat and Casablanca cities for 8 million habitants. Geologically, the Bouregreg basin in located in the Moroccan Central Massif, which mainly consists of Paleozoic formations that were subjected to the Hercynian orogeny.

## 2.2. Datasets

### 2.2.1. Concentration of Suspended Solid Measurements

For the modeling of the concentrations of suspended solids (CSS), we chose four hydrological stations located at the entrance of the SMBA dam reservoir. These hydrological stations are Aguibat Ziar on the Bouregreg River, Ras Fatia on the Grou river, Sidi Mohammed Cherif on the Mechraa River and Ain Loudah on the Korifla River. At all of the stations involved, we carried out daily monitoring of the CSS. These stations control a basin of approximately 87% of the total catchment area of the SMBA dam. In addition, samples were taken daily during low water periods and hourly during high water periods. For each sample, the date, time, and scale values were noted on the bottle. The samples were then analyzed in the laboratory, filtered under vacuum using filtering membranes (0.45 μm), and weighed. Table 1 summarizes the number of daily data samples used at the four hydrologic stations.

**Table 1.** Statistical characteristics of CSS samples at 4 hydrological stations studied.

| Name of Hydrological Station | River | Basin Area (km²) | N°IRE ABHBC Code | Period of Observation Considered | Number of CSS Sample | Mean Period (g/L) | Max Period (g/L) | Min Period (g/L) | Standard Deviation |
|---|---|---|---|---|---|---|---|---|---|
| Aguibat Ziar | Bouregreg | 3681 | 3118/13 | 1 September 2016 to 31 August 2021 | 1253 | 0.83 | 32.72 | 0.00 | 1.08 |
| Ras Fathia | Grou | 3485 | 989/20 | | 1634 | 1.20 | 86.9 | 0.00 | 1.70 |
| S.M. Cherif | Mechraa | 656 | 2673/20 | | 1662 | 0.63 | 16.89 | 0.00 | 0.76 |
| Ain Loudah | Korifla | 699 | 2674/21 | | 470 | 1.05 | 24.7 | 0.00 | 1.47 |

### 2.2.2. Rainfall and Runoff Measurements

For the prediction of the suspended sediment load at each of the four stations located immediately upstream of the SMBA dam, we used the daily data of the rainfall and the runoff of the rivers measured at the hydrological stations located at the level of each sub-basin controlled by one of the 4 stations studied (Tables 2 and 3). These parameters were selected as they are the main influencing factors in erosion process [8,9]. The following tables give the distribution and characteristics of the data by sub-basin of these stations.

**Table 2.** Statistical characteristics of rainfall data by sub-basin during the period from 1 September 2016 to 31 August 2021.

| Name of Sub-Basin | Rainfall Station | N°IRE Code | Mean (mm) | Max (mm) | Min (mm) | Standard Deviation |
|---|---|---|---|---|---|---|
| Bouregreg basin at Aguibat Ziar | Aguibat Ziar | 3118/13 | 1.3 | 122.5 | 0.0 | 2.23 |
| | Lalla Chafia | | 0.9 | 42.9 | 0.0 | 1.64 |
| | Sidi Amar | | 1.1 | 50.0 | 0.0 | 1.94 |
| | Tslat | | 1.3 | 60.0 | 0.0 | 2.11 |
| Grou at Ras Fathia | Ras Fathia | 989/20 | 1.0 | 41.2 | 0.0 | 1.81 |
| | Sidi Jabeur | | 0.8 | 44.5 | 0.0 | 1.50 |
| | Ouljat Haboub | | 0.8 | 39.8 | 0.0 | 1.38 |
| Korefla basin at Ain Loudah | Ain Loudah | 2674/21 | 0.9 | 59.6 | 0.0 | 1.57 |
| Korefla basin at S.M. Cherif | S.M. Cherif | 2673/20 | 0.9 | 54.7 | 0.0 | 1.56 |

**Table 3.** Statistical characteristics of runoff data by sub-basin for the period from 1 September 2016 to 31 August 2021.

| Name of Sub-Basin | Rainfall Station | N°IRE Code | Mean (m³/s) | Max (m³/s) | Min (m³/s) | Standard Deviation |
|---|---|---|---|---|---|---|
| Bouregreg basin at Aguibat Ziar | Aguibat Ziar | 3118/13 | 4.16 | 199.04 | 0.00 | 5.64 |
| | Lalla Chafia | | 3.06 | 196.70 | 0.00 | 4.85 |
| | Sidi Amar | | 0.48 | 24.04 | 0.00 | 0.61 |
| | Tslat | | 0.81 | 23.50 | 0.00 | 1.04 |
| Grou at Ras Fathia | Ras Fathia | 989/20 | 4.58 | 349.21 | 0.00 | 6.59 |
| | Sidi Jabeur | | 3.74 | 326.54 | 0.00 | 5.29 |
| | Ouljat Haboub | | 3.52 | 262.37 | 0.00 | 5.43 |
| Korefla basin at Ain Loudah | Ain Loudah | 2674/21 | 0.52 | 48.83 | 0.00 | 0.88 |
| Korefla basin at S.M. Cherif | S.M. Cherif | 2673/20 | 0.42 | 38.58 | 0.00 | 0.62 |

*2.3. Methodology*

2.3.1. Machine Learning Models

In the present study, we applied the regression-based ML models, namely: random forest (RF), adaptive boosting (AdaBoost), support vector machine for regression (SVR), k-nearest neighbor (k-NN), and artificial neural network (ANN) models. Further details on the ML-based models can be found in [30–35]. However, the following subsections provide a short description of these models.

Random Forest

Random forest (RF) is a tree-based ML model [36]. Regression and classification are conducted by aggregating a technique that operates by constructing an ensemble of decision trees in training by swapping and changing the covariates to improve the prediction performance. The final output (target) is calculated through the weighted average of tree outputs. Executing this model requires a number of trained trees and an amount of the variable used in each tree [37]. Indeed, these parameters have an important role in the model stability and, therefore, in its prediction accuracy [38].

AdaBoost

Adaboost is an ensemble ML model developed by Freund and Schapire (1997) [33]. It can be applied either for classification or regression purpose. Adaptive boosting (AdaBoost) is a tree-based ensemble ML model [39,40]. Recently, this approach appeared to be an efficient regression model in environmental sciences, namely, for regression and data-based augmentation techniques [15,41].

For the data, $S = \{(x_i, y_i), i = 1, 2, 3, \dots, N\}$, where each $x_i$ is in some instance $X$ and each $y_i$ is in some target (output) $Y$, and, for a series of rounds ($M$), the algorithm initializes the distribution ($D$) (or weight) as follows:

$$D_i^1 = \frac{1}{N} \quad for\ i = \{1, \dots . N\} \tag{1}$$

Then, for $j = 1$ to $M$, Adaboost algorithm builds weak models $h_j$ from the training dataset using $D$, which minimizes $\varepsilon_j$ and satisfies $\varepsilon_j < 0.5$ conditions.

$\varepsilon_j$ is a weighted error of the $j$th model and is given by Equation (2).

$$\varepsilon_j = \sum_{i:h_j\ (x_i) \neq y_i} D_i^j \tag{2}$$

The weight "confidence" $\alpha_j$ of the $j$th model is calculated by Equation (3).

$$\alpha_j = \frac{1}{2} \ln\left(\frac{1 - \varepsilon_j}{\varepsilon_j}\right) \tag{3}$$

The distributions for next iteration were updated as follows:

$$D_i^{j+1} = e^{-y_i h_j(x_i)\alpha_j} D_i^j \tag{4}$$

$$D_i^{j+1} = \frac{D_i^{j+1}}{\sum_{i=1}^{N} D_i^{j+1}} \tag{5}$$

The prediction for new dataset was conducted by combining weighted majority vote of the models $h_j$.

$$H(x') = sign[\sum_j^M \alpha_j h_j(x')] \quad (whithout\ sign\ for\ the\ regression) \tag{6}$$

Support Vector Machine (SVM)

Support vector machine (SVM) is a discriminative technique that was introduced for the first time by Vapnik (1995) [42]. It is based on a hyper-plane in order to minimize the error and the kernel function, such as radial basis function [43], sigmoid, linear kernel, and polynomial function. This method has demonstrated a high accuracy prediction for several regression applications [24].

For modeling system ($S$) with observation dataset ($D_s$), $D_s = \{(x_i, y_i)\}_{i=1}^{n}$, where $x_i$ represents the inputs and $y_i$ the outputs, with a linear function, as shown in Equation (7).

$$f(x) = \langle \omega \times \varnothing(x) + b \rangle \tag{7}$$

The optimal function is the minimization of Function (8) (subject to Equation (9)). Hence, the loss functions, such as $\epsilon$-insensitive, quadratic, and Hubber methods, can be used.

$$\min(\omega, b, \xi^-, \xi^+) = \frac{1}{2} \times \left\| \omega^2 \right\| + C \times \sum_{i=1}^{n} (\xi i^- + \xi i^+) \tag{8}$$

$$Subject\ to \begin{cases} y_i - \omega^T \times \varnothing(x) - b \leq \varepsilon + \xi i^- \\ -y_i + \omega^T \times \varnothing(x) + b \leq \varepsilon + \xi i^+ \\ \xi i^-,\ \xi i^+ \geq 0 \\ i = 1, 2 \ldots .. n \end{cases} \tag{9}$$

where $\varnothing(x)$ is a Kernel function ($k$), such as polynomial, radial basis, and linear functions; $\omega$ and $b$ represent weigh and basis vectors; $C$ is a pre-specified value to penalize the training error; and $\xi i^-$ and $\xi i^+$ are the lower and upper constraints on the output.

This study adopted the radial basis function (RBF) given by Equation (16) as kernel function [43].

$$k(x_i, x_j) = exp^{(-\gamma |x_i - x_j|^2)} \tag{10}$$

Artificial Neural Network (ANN)

ANN models are constructed by three layer types, namely: input layer, hidden layers (HL), and the output layer [44]. They are interconnected through neurons, which are characterized by weight and bias. The weighted input variables summed with the bias of the layer are transformed from the $j$th layer to the $(j + 1)$th layer by transfer function ($f$), and so on, until the output [44]. The training phase is repeated by changing the weights and the biases of the layers until good prediction accuracy (root mean square error) is achieved. To simplify this method, let us take a simple model with one HL. The outputs ($Y_k$) are given by the following equation [45]:

$$Y_k = f_k(\sum_{i=1}^{m} W_{jk} \times f_j(\sum_{i=1}^{n} X_i W_{ij})) + W_0 \tag{11}$$

where $n$ is the input variable numbers, $m$ is the neurons in the HL, $p$ is the neurons of the output layer, $W_0$ is the bias, and $W_{jk}$ and $W_{ij}$ are the weights between the $j$th neuron and the $k$th output neuron and between the $i$th neuron and $j$th neuron, respectively, whereas $f_k$ and $f_j$ are the transfer functions of the neurons $k$ and $j$ of the output and hidden layers, respectively.

k-Nearest Neighbor (k-NN)

The k-NN algorithm is a memory-based method (non-parametric method); the predicted values are estimated based on the information on the neighboring observed ones.

The estimated values are obtained by the average of the nearest $k$ observed values and give more weight to the nearer ones, as shown in Equations (12) and (13).

$$\overline{f}\left(\overrightarrow{x}_0\right) = \sum_{i=1}^{k} \beth_{i0} f\left(\overrightarrow{x}_i\right) \tag{12}$$

$\overline{f}\left(\overrightarrow{x}_0\right)$ is the predicted response, $\overrightarrow{x}_0$ is the vector of independent inputs, $f(\overrightarrow{x}_i)$ is the observed response, $\overrightarrow{x}_i$ is the vector of the nearest $k$ observed values, and $\beth_{i0}$ is the weight between the $\overrightarrow{x}_0$ and $\overrightarrow{x}_i$, and is given by Equation (13).

$$\beth_{i0} = \frac{\sum_{i=1}^{k} \left|\overrightarrow{x}_i - \overrightarrow{x}_0\right|}{\left|\overrightarrow{x}_i - \overrightarrow{x}_0\right|} \tag{13}$$

As shown in this equation, the number $k$ is important for the model performances. It should be noted that there are others distance functions such as Euclidian, Shebyshev, and Mahalanobis.

### 2.3.2. NARX Input Method

In this study, we proposed NARX-based modeling to incorporate the delay response of the suspended sediment load to the hydrological events. Considering $N$ of datasets related to the system ($S$) to be modeled using ML algorithms, $S = \{(X_i, y_i), i = 1, 2, \ldots, N\}$, where $X_i = [x_{i,1}, x_{i,2}, x_{i,3} \ldots x_{i,p}]^T$ is the input matrixes ($p$ is the input numbers), $y_i = [y_{i,1}, y_{i,2}, y_{i,3} \ldots y_{i,q}]^T$ is the output vector, and $i$ is the time instant. Hence, the NARX-based ML algorithms for predicting the SSL have NARX input variables as follows:

$$y_i = MLf_m(\varphi(i), \theta) \tag{14}$$

where $\varphi(i) = (x_i, x_{i-1}, x_{i-2}, \ldots, x_{i-lx})$ is the regression vector, $\theta$ is the parameter vector, $MLf_m$ is a unknown ML model function, and $lx$ and $ly$ denote the lag numbers of the input $x$ and output $y$, respectively. Therefore, the final input matrixes $X_i^T$ and the output vector $y_i^T$ for training of the ML models are as follows:

$$\begin{cases} X^T = [x_i, x_{i-1}, x_{i-2}, \ldots, x_{i-lx}] \\ y^T = y_i \\ \max(lx) < i \leq N \end{cases} \tag{15}$$

The unknown function $MLf_m$ of the models includes, as examples, the hyper-parameter, loss functions, and model structures depending on the model. For instance, number of the hidden layers and function transfer for ANN model, and kernel function and penalty parameter ($C$) for SVR model. However, the $MLf_m$ and lag numbers are identified during the tuning process using trial-error procedures.

In this study, we used Python language embedded in Anaconda platform. Pandas, Matplotlib, and Scikit-learn libraries were imported and used. Pandas and Matplotlib are the libraries that were used for dataset loading and visualization processes, respectively. Meanwhile, Scikit-learn library was used for implementing regressor ML models.

### 2.3.3. Model Evaluation Metrics

To evaluate prediction model performances, several indices and metrics can be used, such as coefficient of correlation, root mean square error (*RMSE*), and Nash–Sutcliffe-efficiency.

$$RMSE = \sqrt{\frac{\sum(Pi - Oi)^2}{n}} \tag{16}$$

$$NSE = 1 - \frac{\sum_{i=1}^n (Pi - Oi)^2}{\sum_{i=1}^n (Pi - \overline{O})^2} \tag{17}$$

$$r = \left( \frac{\sum_{i=1}^n (Oi - \overline{O})(Pi - \overline{P})}{\left[ \sum_{i=1}^n (Oi - \overline{O})^2 \sum_{i=1}^n (Pi - \overline{P})^2 \right]^{0.5}} \right) \tag{18}$$

*Oi* and *Pi* are observed and model-simulated values, respectively; $\overline{O}$ represents the mean of observed values; and *n* = number of the observations used.

However, in this study, we used the generalization ability as an additional metric to evaluate the models. This metric was defined by [46] through the following equation:

$$GA = \frac{RMSE\ (validation)}{RMSE\ (training)} \tag{19}$$

However, to rank the models for simulation purposes, EL Bilali et al. [47] suggested the classification of the machine learning models according to categories of perfect, excellent, good, and poor models in terms of *GA* as follows:

- If *GA* = 1, model is perfect;
- If $0.75 \leq GA < 1$ or $1 < GA \leq 1.35$, the model is excellent;
- If $1.35 < GA \leq 2$ or $0.5 \leq GA < 0.75$, the model is good;
- If *GA* > 2 or *GA* < 0.5, the model is poor and considered unsuitable for simulation purposes.

## 3. Results

### 3.1. Exploratory Data Analysis (EDA)

In developing ML models, the selection of the feature variables is a keystone process in the improvement of the prediction accuracy and the generalization ability of the models. Despite being black box models that do not explain the mechanism involved in the process to be modeled, selecting the causal input variables for training ML models is required. In this study, we focused on the hydro-climatic variables in order to predict the suspended sediment load (SSL), namely: rainfall in the basins and discharges both upstream and downstream. The principal component analysis was conducted to demonstrate the importance of the selected variables, especially for the basin monitored by several hydrological stations. Figure 2 illustrates the PCA of the datasets at the Bouregreg basin (Aguibat Ziar) and Grou basin (Ras Fathia Station. The PCA of Bouregreg datasets (Aguibat Ziar) revealed that there is 74.4% of information regarding the total variance, of which, 46.9% is explained by PC1 and 27.5% by PC2 (Figure 2a). As for the Grou basin, the PCA showed that 69.4% of information on the total variance is explained by the PC1 (46.6%) and PC2 (23.0%). These results demonstrate the importance of the input variables. However, the antecedent values of these variables can impact the suspended sediment load, as the soil erosion does not only depend on the current conditions but also on the initial conditions. For this reason, using NARX input variables is suggested to improve the prediction accuracy of the models.

A Pearson correlation-based analysis of the dataset was carried out to explore the potential relationship between the variables, including antecedent values. Figure 3 presents the matrix correlation of the variables in the studied basins. It was observed that, except for the discharge at Ras Fathia stations and Sidi Jabeur stations, all other variables are not highly inter-correlated. Such results demonstrate that the selected variables are not redundant in the prediction of the SSL. Interestingly, except for some cases, the antecedent variables of rainfall and discharge at (day-1) and (day-2) are more correlated than the actual variable. For instance, the actual variables at the basin outlets are more important than those of antecedent days. This is due to the delayed SSL response to the hydrological events of the basin. Consequently, embedding the antecedent hydroclimatic conditions as input variables is suggested to improve the prediction accuracy of the models.

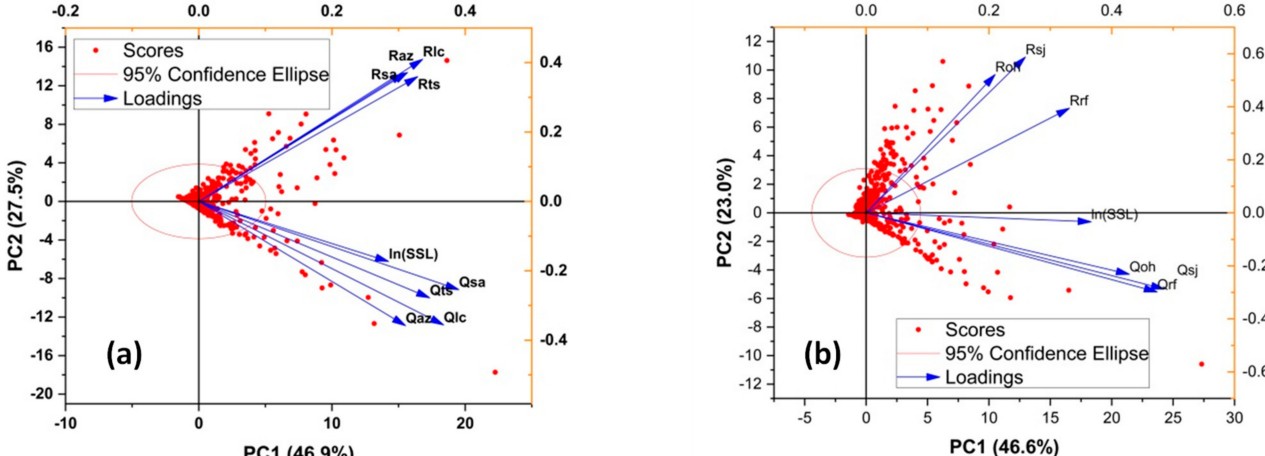

**Figure 2.** Principal component analysis (PCA). (**a**): PCA of data recorded at Bouregreg basin; ln(SSL) is the Napierian logarithm of the suspended solid; Ras and Qas are rainfall and discharge at Aguibat Ziar station; Rlc and Qlc are rainfall and discharge at Lala Chafia station; Rts and Qts are the rainfall and discharge at Tsalat station; Rsa and Qsa are the rainfall and discharge at Sidi Amar station. (**b**): PCA of dataset recorded at Grou basin.

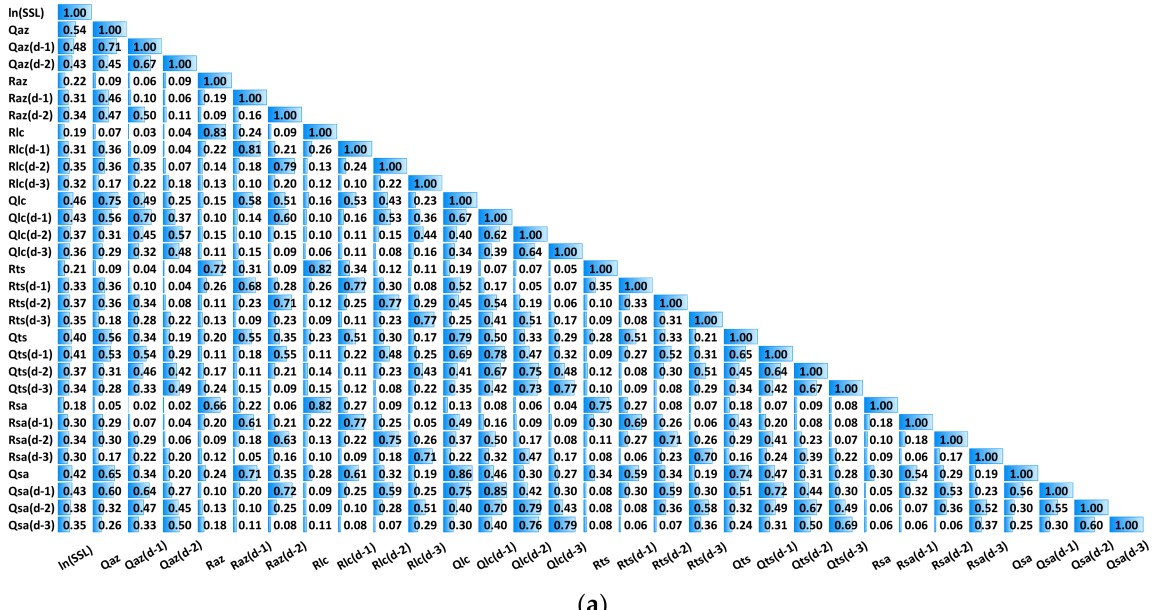

(**a**)

**Figure 3.** *Cont.*

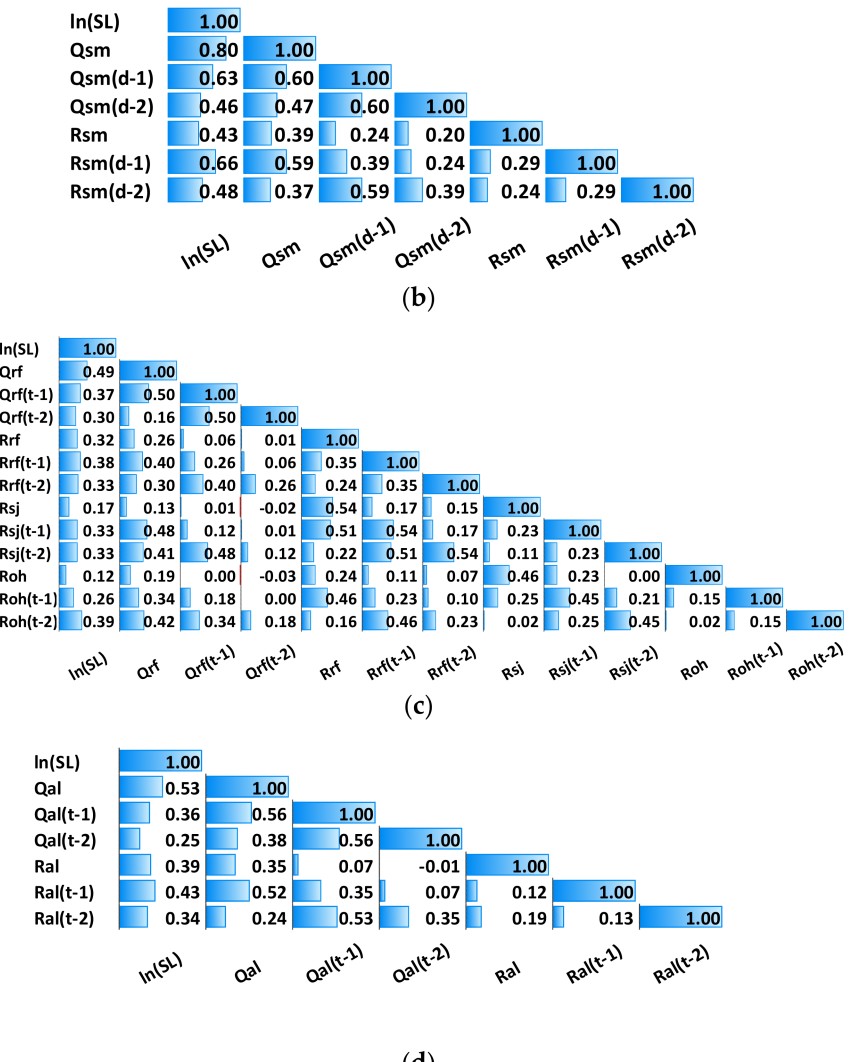

**Figure 3.** Correlation matrix of the variables. (**a**): Correlation matrix of the variables at Aguibat Ziar (az) station; (**b**): correlation matrix of the variables at Sidi Mohammed Cherif (sm) station; (**c**) correlation matrix of the variables at Ras Fathia (rf) station (sm); (**d**): correlation matrix of the variables at Ain Loudah (al) station. The Napierian logarithm of the suspended sediment load ln(SSL) is positively correlated with discharges and rainfalls in the basin. The rainfalls and flow discharge at (day-i) are more important.

### 3.2. Training Process of the Machine Learning Models

In this step, five ML models with NARX input variables, namely random forest, AdaBoost, support vector machine (SVM), k-nearest neighbor (kNN), and artificial neural network (ANN), were trained, tuned, and evaluated using cross-validation with a number of folds (k = 10) to improve the prediction accuracy. The tuning process includes changing the hyper-parameter and transfer functions of the models. However, Table 4 presents the adopted parameters in this study.

Table 5 presents the ML performance during the training phase. This table clearly shows that all models have a high potential accuracy in predicting the daily suspended solid (SSL) in four main basins in the Great Bouregreg basin (BW), justified by the fact that the NSE ranges from 0.57 to 0.84 in the Aguibat Ziar station, from 0.61 to 0.8 in the Ras Fathia station, from 0.41 to 0.62 for the SM Cherif station, and from 0.66 to 0.76 in the Ain Loudah station. It was observed that the random forest and AdaBoost models outperformed the other models during the training phase.

**Table 4.** Adopted hyper-parameter and functions of the trained models.

| Model | Parameters/Functions/Algorithm |
|---|---|
| RF | 50 Trees |
| AdaBoost | Loss function: exponential<br>learning rate: 0.5<br>Estimator number: 250 |
| SVR | $C = 350$<br>Kernel function: RBF ($\gamma = 1.3$)<br>$\varepsilon$-function loss, $\varepsilon = 0.002$ |
| k-NN | k = 5<br>Distance function: Euclidian |
| ANN | 3 layers<br>12 neurons in hidden layer<br>Algorithm: Levenberg<br>Function activation: Sigmoid<br>Identity in output layer<br>Epoch number: 1000<br>Learning rate: 0.01<br>Momentum coefficient: 0.85 |

Notes: RF: random forest; AdaBoost: adaptive boosting; SVR: support vector regression; k-NN: k-nearest neighbor; ANN: artificial neural network.

**Table 5.** Machine learning model performances during the training phase.

| Models | *r* | *RMSE* | *NSE* ML | Musle_Calibrated |
|---|---|---|---|---|
| **Aguibat Ziar** | | | | |
| Random Forest | 0.92 | 1.18 | 0.84 | |
| AdaBoost | 0.92 | 1.20 | 0.84 | |
| SVM | 0.76 | 1.95 | 0.57 | 0.46 |
| kNN | 0.88 | 1.49 | 0.75 | |
| Neural Network | 0.79 | 1.83 | 0.62 | |
| **Ras Fathia** | | | | |
| Random Forest | 0.89 | 1.50 | 0.80 | |
| AdaBoost | 0.89 | 1.54 | 0.78 | |
| SVM | 0.78 | 2.08 | 0.61 | 0.02 |
| kNN | 0.85 | 1.83 | 0.70 | |
| Neural Network | 0.82 | 1.90 | 0.67 | |
| **SM Cherif** | | | | |
| Random Forest | 0.79 | 1.62 | 0.62 | |
| AdaBoost | 0.76 | 1.74 | 0.56 | |
| SVM | 0.73 | 1.79 | 0.54 | 0.47 |
| kNN | 0.68 | 2.01 | 0.41 | |
| Neural Network | 0.76 | 1.70 | 0.58 | |
| **Ain Loudah** | | | | |
| Random Forest | 0.87 | 1.91 | 0.76 | |
| AdaBoost | 0.84 | 2.13 | 0.70 | |
| SVM | 0.81 | 2.26 | 0.66 | 0.30 |
| kNN | 0.83 | 2.26 | 0.66 | |
| Neural Network | 0.84 | 2.12 | 0.70 | |

*3.3. Validation of the ML Models*

This step was carried out to evaluate whether the trained ML models are generalizable in order to predict the suspended sediment load for the dataset unseen during the training phase. To that end, we simulated the SSL recorded during the year 2021 and evaluated the model performances. Table 6 presents the model performances during the validation phase. It was observed that, except for the kNN model in predicting the SSL in the Ain Loudah station, most of the models presented good accuracy during the validation phase, with a

NSE ranging from 0.77 to 0.55, an r ranging from 0.82 to 0.92, and a RMSE ranging from 1.34 to 2.96 $\ln(t \cdot d^{-1})$.

**Table 6.** Machine learning model performances during the validation phase.

| Models | r | RMSEln ($t \cdot d^{-1}$) | NSE |
|---|---|---|---|
| **Aguibat Ziar** | | | |
| RF | 0.84 | 1.52 | 0.68 |
| AdaBoost | 0.84 | 1.51 | 0.68 |
| SVM | 0.79 | 1.68 | 0.61 |
| kNN | 0.78 | 1.81 | 0.55 |
| ANN | 0.80 | 1.61 | 0.64 |
| **Ras Fathia** | | | |
| RF | 0.90 | 2.27 | 0.70 |
| AdaBoost | 0.88 | 2.38 | 0.67 |
| SVM | 0.90 | 2.35 | 0.68 |
| kNN | 0.82 | 2.56 | 0.62 |
| ANN | 0.92 | 1.99 | 0.77 |
| **SM Cherif** | | | |
| RF | 0.86 | 1.53 | 0.74 |
| AdaBoost | 0.84 | 1.66 | 0.69 |
| SVM | 0.85 | 1.69 | 0.69 |
| kNN | 0.82 | 1.75 | 0.66 |
| ANN | 0.91 | 1.34 | 0.80 |
| **Ain Loudah** | | | |
| RF | 0.90 | 2.60 | 0.64 |
| AdaBoost | 0.84 | 2.76 | 0.61 |
| SVM | 0.84 | 2.96 | 0.55 |
| kNN | 0.78 | 3.19 | 0.47 |
| ANN | 0.86 | 2.72 | 0.61 |

To compare the trained models for simulating the SSL recorded during the 2021 year, Taylor diagrams were plotted, and are presented in Figure 4. It shows that the AdaBoost and RF models outperform the other models in predicting the SSL in Aguibat Ziar. The ANN model is more accurate than the other ones for simulating the SSL in the Ras Fathia and SM Cherif stations. As for the Ain Loudah station, the RF model is more accurate, followed by the ANN, RF, and SVM models. However, the kNN model has the lowest accuracy in predicting the SSL in all stations in comparison to the other models. However, according to the suggestions in [48], it has fairly acceptable performances (NSE > 0.5) for the SM Cherif, Ras Fathia, and Aguibat Ziar stations (Table 6).

Besides, Figure 5 illustrates the boxplot and the fitted normal distribution of the model errors in simulating the SSL recoded during 2021. This figure demonstrates that the model errors obey a Gaussian distribution, as they are evenly distributed over small values. Figure 5a shows that the SVM model has a narrow box in comparison to the other models in simulating the SSL in Aguibat Ziar, and even its NSE is lower than those of the Adaboost, RF, and ANN models. This is justified by the fact that the NSE is sensitive to the outlier values (Figure 5a). A similar result was observed for the kNN and SVM models in predicting the SSL at the SM Cherif station (Figure 5c). The normal distribution of the model errors demonstrates their stability in simulating the SSL for the dataset unseen during the training process.

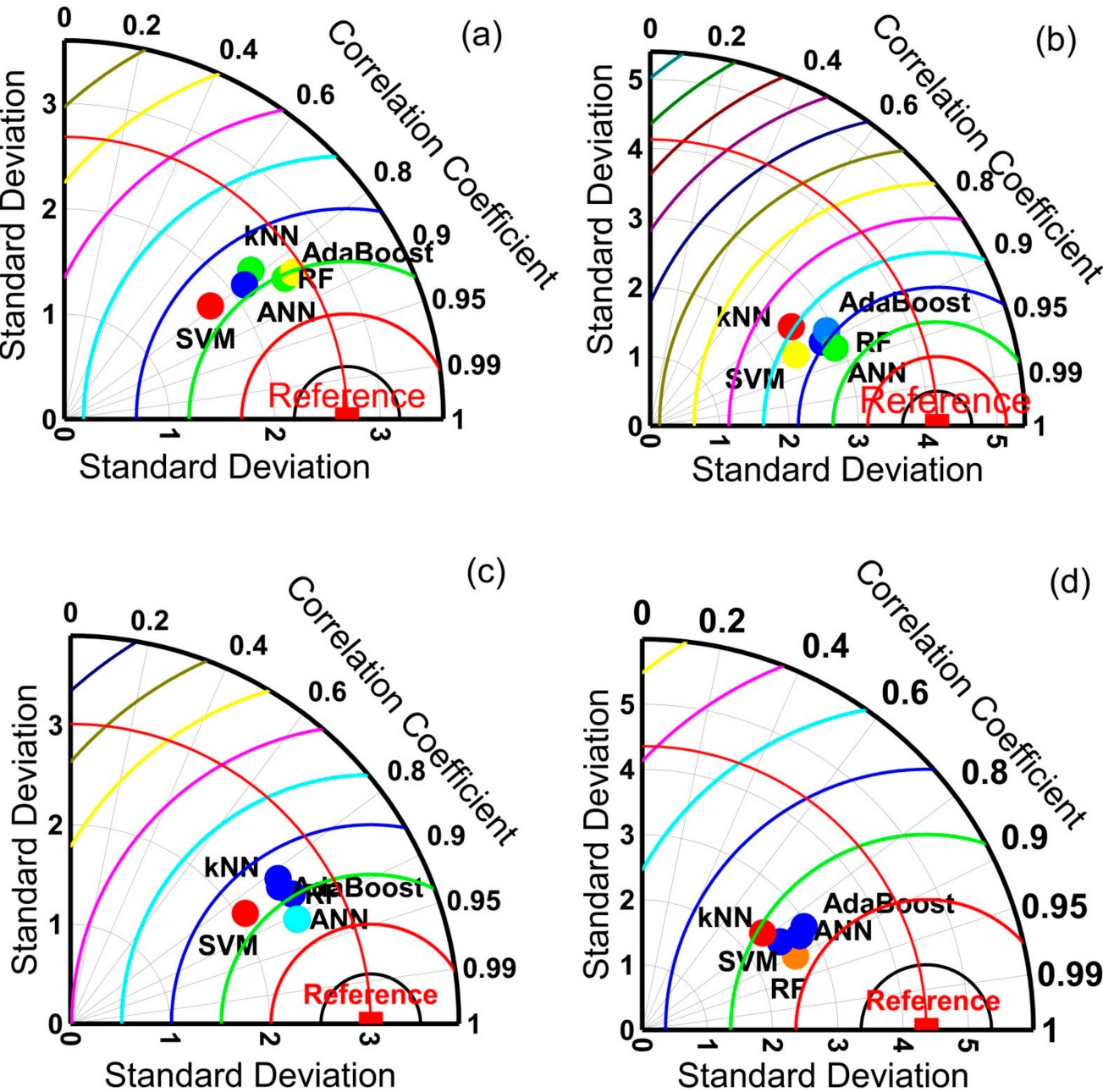

**Figure 4.** Taylor diagrams for comparing trained machine learning models, namely random forest (RF), adaptive boosting (AdaBoost), artificial neural network (ANN), and k-nearest neighbor (kNN). (**a**): Comparison of the ML models for predicting suspended sediment load at Aguibat Ziar station; (**b**): comparison of the ML models for predicting SSL at Ras Fathia station; (**c**): comparison of the ML models at SM Cherif station; (**d**): comparison of the ML models at Ain Loudah station.

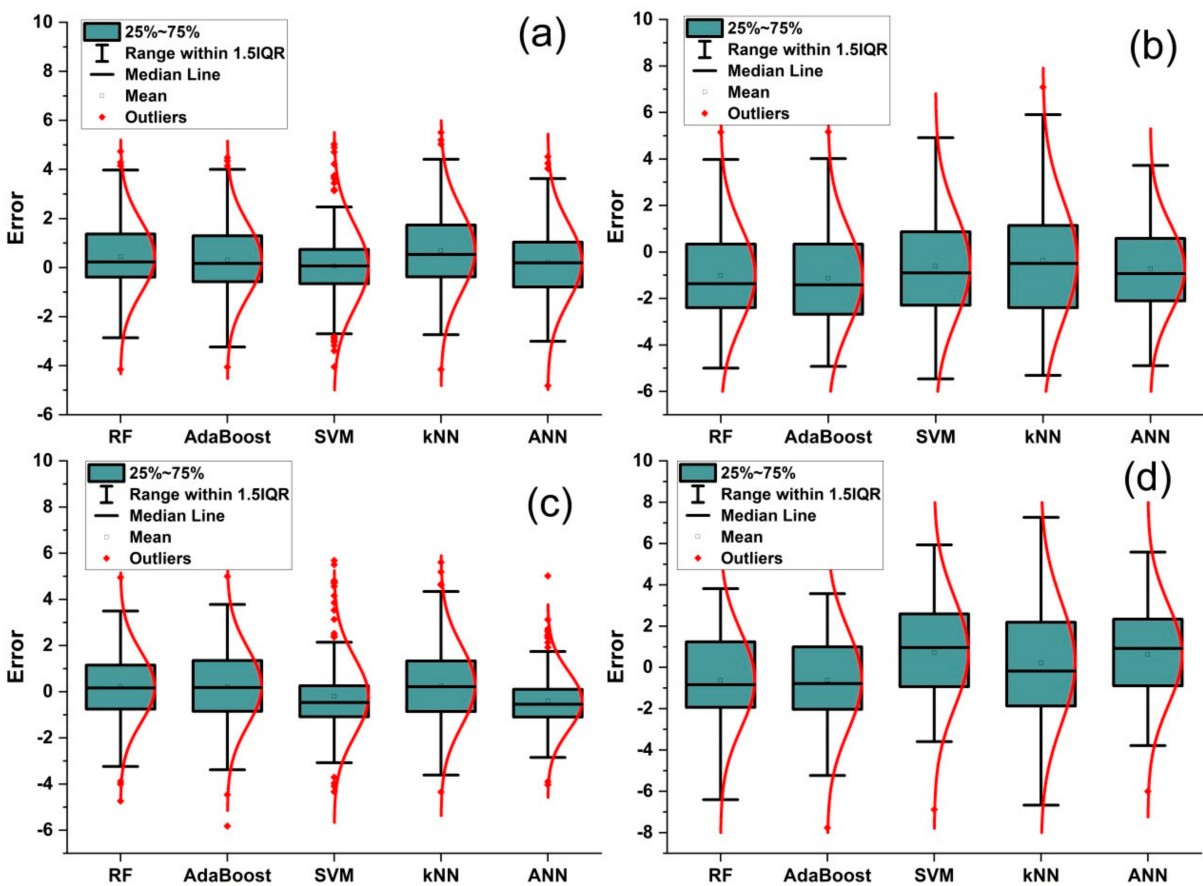

**Figure 5.** Distribution of the model errors during the simulation of the suspended sediment load recorded in 2021, namely random forest (RF), adaptive boosting (AdaBoost), artificial neural network (ANN), and k-nearest neighbor (kNN). Red lines represent the fitted normal distributions of the error, and points outside the box plots are the outlier values of the errors. (**a**): at Aguibat Ziar station; (**b**): at Ras Fathia station; (**c**): at SM Cherif station; (**d**): at Ain Loudah station.

### 3.4. Generalization Ability and Uncertainties

In this study, the uncertainty analysis and classification of the developed models were conducted, and are presented in Table 7. From this table, it was observed that most of the models have a GA that is good to excellent, demonstrating the good training of these models and, therefore, their utility for simulating the SSL at the studied basin. Besides, the confidence intervals with a significance level of 95% (z-score = 1.96) of the models were computed, and are presented in Table 7.

**Table 7.** Generalization ability, lower limit errors, and upper limit errors of the models.

| Basin | | RF | AdaBoost | SVM | kNN | ANN |
|---|---|---|---|---|---|---|
| | GA | 1.29 | 1.26 | 0.86 | 1.21 | 0.88 |
| Aguibat Ziar | LL 95% | −2.88 | −2.93 | −3.71 | −3.88 | −3.50 |
| | UL 95% | 2.73 | 2.80 | 3.78 | 2.87 | 3.71 |
| | GA | 1.52 | 1.54 | 1.13 | 1.40 | 1.04 |
| Ras Fathia | LL 95% | −3.02 | −3.05 | −4.11 | −3.92 | −3.68 |
| | UL 95% | 2.85 | 3.00 | 4.03 | 3.02 | 3.78 |
| | GA | 0.94 | 0.96 | 0.94 | 0.87 | 0.79 |
| SM Cherif | LL 95% | −3.22 | −3.20 | −3.55 | −4.35 | −3.35 |
| | UL 95% | 3.14 | 3.57 | 3.45 | 3.26 | 3.33 |
| | GA | 1.36 | 1.30 | 1.31 | 1.41 | 1.28 |
| Ain Loudah | LL 95% | −3.88 | −4.26 | −4.47 | −4.89 | −4.18 |
| | UL 95% | 3.59 | 4.08 | 4.41 | 3.65 | 4.16 |

Notes: GA: generalization ability; LL 95%: lower limit with 95% level confidence; UL 95%: upper limit with 95% level confidence; SVM: support vector machine; kNN: k-nearest neighbor; Adaboost: adaptive boosting.

Based on Table 6 and Figure 4, the best models for predicting the SSL in the Aguibat Ziar, Ras Fathia, SM Cherif, and Ain Loudah stations are Adaboost, ANN, and random forest, respectively. Figure 6 presents a graphical comparison of the observed and simulated ln(SSL) with regard to the UL and LL errors provided by these models. This figure shows that, except for isolated cases, most of the simulated ln(SSL) values are included in the confidence interval with a level of 95%. More importantly, the models are capable of predicting the SSL peaks, as shown in Figure 6.

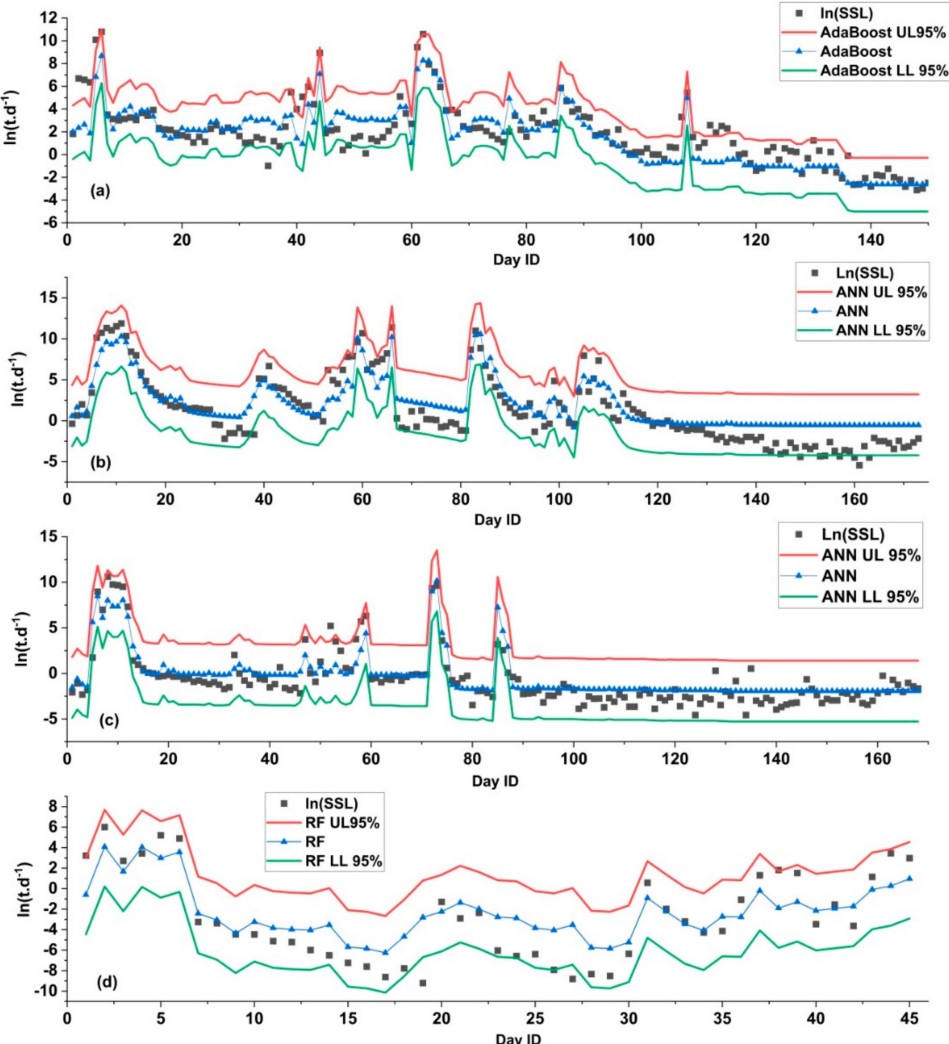

**Figure 6.** Superimposed plots of the measured and simulated ln(SSL) with 95% upper and lower limits using the best fitted ML models, namely: (**a**) adaptive boosting (Adaboost) model for simulating the SSL in Aguibat Ziar; (**b**) artificial neural network (ANN) model for simulating the SSL in Ras Fathia station; (**c**) artificial neural network (ANN) model for simulating the SSL in SM Cherif station; (**d**) random forest (RF) model for simulating the SSL in Ain Loudah station.

## 4. Discussion

Sediment transport at the basin's scale is a highly dynamic phenomenon and depends on a multitude of combination variables that make its estimation a difficult task. Indeed, several process-based models appeared to be valuable tools to predict the sediment load at the basin's scale. Meanwhile, this approach requires the monitoring of several physical parameters for calibration and validation processes that affect the accuracy of the developed model, especially in poorly monitored areas. For instance, in the Bouregreg basin, the MUSLE model had a NSE less than 0.5 in the prediction of the sediment yield in the Aguibat Ziar, Ras Fathia, SM Cherif, and Ain Loudah stations [28]. Interestingly, ref. [48] suggested

that, in order to be acceptable for simulating the sediment load, it should have a NSE > 0.5. Therefore, to overcome this challenge, this study proposed ML-based modeling to improve the prediction accuracy in predicting the suspended sediment load in the Bourereg basin.

The evaluation of the model performance during the simulation of the SSL recorded in 2021 showed that most ML models (except for kNN in the Ain Loudah station) presented a NSE over 0.5, indicating their suitability for simulation purposes. kNN and SVM models had a lower performance for simulating the SSL in all studied stations. This is in agreement with previous published studies, such as Wang et al. [49] and El Bilali et al. [14], where their findings demonstrated that the tree-based and ANN models are more accurate than SVM and k-NN ones. Regarding the SSL prediction, ref. [50] found that the data-driven techniques can efficiently predict the SSL, with a NSE of 0.90, in China. Indeed, the authors used several inputs variables, including the antecedent SSL. Meanwhile, including the antecedent SSL as an input variable to predict the actual SSL can reduce the practical implication of the models, since this variable needs measurement. In contrast, using only hydro-climatic variables as input features in this study was enough to accurately predict the SSL. More importantly, the generalization ability (GA) of the ML models in predicting the SSL is crucial for the selection of the suitable model. This study evaluated the models according to the performances, in addition to the GA, to assess whether the models keep the accuracy during the validation process, and the variation of this accuracy, especially when the models were over-fitted or under-trained. The results showed that the models were ranked.

The applied methodology was an accurate tool to predict the daily SSL in immediate upstream stations of the SMBA reservoir, which could be integrated as a powerful module in a decision support system (DSS) of water resource planning and management in the studied area in order to evaluate the reservoir sedimentation with a high frequency. Indeed, the suspended sediment load from the intermediate basin and the erosion of the banks of the SMBA reservoir were not considered in the present study, which can be among its limitations.

Meanwhile, coupling the proposed ML models with a stream-based model to evaluate the sediment inflow into the SMBA reservoir is suggested to improve the accuracy, since the studied stations are upstream of the reservoir [51]. Since the stream-based models require too many physical datasets pertaining to the river reach, such as the digital elevation model (DEM), Manning–Strickler coefficient, topographical surveys of the bridges and culverts, and others, the availability of the bathymetric survey dataset could be valuable for statistically linking the cumulative SSL and the cumulative reservoir sedimentation. Hence, it will be a reliable alternative to the previous suggestion. Additionally, the evaluation of the erosion of the SMBA reservoir banks using ML models requires bathymetric measurement datasets with enough frequency, and can be suggested to improve the SMBA reservoir sedimentation monitoring.

Besides, the sedimentation can be managed both at the reservoir scale and at the basin scale. At the reservoir scale, the flushing process through the water outflow rate and the dredging techniques are the main operations used to manage/reduce the reservoir sedimentation [52], whereas, at the basin scale, the best practice management (BPM) are outstanding methods to mitigate the soil erosion. Despite being an accurate approach to estimate the SSL upstream of the SMBA reservoir, the proposed ML models are not capable of simulating the BPM at the basin scale, because the input features do not include the physical parameters related to the basin, such as the land cover, soil type, and anthropogenic activities related to the basin to be studied. However, this limitation could be overcome by coupling ML models and process-based models (the soil water assessment tool model, for instance); therefore, it can be suggested for future works. Interestingly, spatial machine learning models embedded in GIS tools could be a fruitful approach to simulate the BPMs in order to reduce the soil erosion over basins.

Eventually, it must be reminded that all of the results of the modelling process are based on daily measurements of suspended sediment loads in the streams of all of the

tributaries of the SMBA dam, performed by the River Basin Agency. Ground observations are indispensable, prior to any modeling process for future predictions, in order to improve the river and dam management.

## 5. Conclusions

An accurate estimation of the suspended sediment load (SSL) in the basins is considered to be a crucial challenge in water resource planning and management. The response of the SSL in basins is nonlinear and highly dynamic, especially in semi-arid regions. In this study, five machine learning models with the NARX input method were applied and evaluated. The mains conclusions of the present study are as follows:

1.  Empowering machine learning by adopting the NARX hydro-climatic input variable method is valuable in the prediction of the daily suspended sediment load;
2.  The artificial neural network is accurate in simulating the SSL in two basins out of four, followed by AdaBoost and random forest models;
3.  The generalization ability metric demonstrated the stability of the applied models in predicting the SSL.

In summary, the proposed methods are powerful tools to overcome the scarce physical dataset required by the process-based model to predict the SSL. However, the continuous monitoring of the SSL in the studied basin will increase the amount of the datasets and, therefore, exploration of the DNN is suggested for future work.

**Author Contributions:** Conceptualization, M.A.E. and A.E.B.; methodology, M.A.E. and A.E.B.; software, M.A.E. and A.E.B.; validation, G.M. and I.K.; formal analysis, M.A.E. and A.E.B.; investigation M.A.E. and A.E.B.; resources, A.Z.; data curation, M.A.E. and A.N.; writing—original draft preparation, M.A.E. and A.E.B.; writing—review and editing, G.M. and I.K.; visualization, M.A.E. and A.E.B.; supervision, G.M. All authors have read and agreed to the published version of the manuscript.

**Funding:** This research received no external funding.

**Data Availability Statement:** All data used are available upon reasonable request.

**Conflicts of Interest:** The authors declare no conflict of interest.

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
