# Peer review of "Predicting Daily Suspended Sediment Load Using Machine Learning and NARX Hydro-Climatic Inputs in Semi-Arid Environment"

_water, doi:10.3390/w14060862_

Round 1

Author Response

General comment to the reviewers

The authors thank the reviewers for their comments and suggestion. We were pleased to take into consideration all comments and suggestions.

Please find bellow answers, comments and more clarifications regarding the present research paper:

Reviwer1

Comment 1: The authors should explain all applied models Random Forest (RF), Adaptive
Boosting (AdaBoost), Support Vector Machine for Regression (SVR), k-Nearest Neighbor
(k-NN), and Artificial Neural Network (ANN) separately and specified the initial parameter
setting of each algorithm.

Response: The applied models were described in the revised manuscript from line 141 to line 206.

Comment 2: Parameter setting is one of the important steps in machine learning-based
modeling that has not been carefully addressed in this manuscript. The authors need to clearly
discuss this issue in their manuscript.

Response: the hyper parameters and functions adopted in the study were provided in Table 4.
Comment 3: The authors used RMSE, R2, and NSE to evaluate the model's performance, but
they didn’t write the equation for these indicators in the model evaluation metrics section.

Response: the equations of the used metrics were added in the revised manuscript accordingly. From line 226 to line 229

Comment 4: A brief review is also needed in the introduction about the application of machinelearning techniques in hydraulic and hydrology, refer to below paper may be useful for theintroduction section:
https://www.sciencedirect.com/science/article/abs/pii/S0022169420302134
https://link.springer.com/article/10.1007/s00521-016-2792-8

Response: the suggestion was followed and references were cited in the introduction part. From line 53 to line 57.
Comment 5: The discussion section should be separated from the conclusion section also; the
conclusion in the revised manuscript should only report the key findings.

Response: The conclusions and Discussions were separated in the revised manuscript

Comment 6: Additionally, it is required to provide some including remarks to further discuss
the proposed method, for example, what are the main advantages and limitations in comparisonwith existing methods?

Response: the advantages and limitations of the applied method were mentioned in the discussion part from line 406 to line 430.

Reviewer 2 Report

In the manuscript “Predicting daily Suspended Sediment Load using Machine Learning and NARX hydro-climatic inputs in semi-arid environment” authors are dealing with important subject because development of models of high accuracy in hydrology has many practical applications.

However, I have some comments:

  1. Authors should put the information about the software used for the development of ML models
  2. In section “2.3.1. Machine Learning with NARX inputs”, Authors mentioned that the MLfm and the lag numbers were identified during the tuning process using trial-error procedures. Authors should present the hyper-parameter of models, and structures as well as transfer functions for ANN models.
  3. In section “2.3.2. Model evaluation metrics”, Authors should present mathematical formula for Nash-Schiff efficiency (NSE). I know Nash-Sutcliffe efficiency as NSE and I’m not sure if it’s the same metrics.

Author Response

General comment to the reviewers

The authors thank the reviewers for their comments and suggestion. We were pleased to take into consideration all comments and suggestions.

Please find bellow answers, comments and more clarifications regarding the present research paper:

Reviwer2

However, I have some comments:

  1. Authors should put the information about the software used for the development of ML models

Response: further information about the implementation was provided in the revised manuscript from line 220 to line 223

  1. In section “2.3.1. Machine Learning with NARX inputs”, Authors mentioned that the MLfm and the lag numbers were identified during the tuning process using trial-error procedures. Authors should present the hyper-parameter of models, and structures as well as transfer functions for ANN models.

Response:Response: the hyper parameters and functions adopted in the study were provided in Table 4.

  1. In section “2.3.2. Model evaluation metrics”, Authors should present mathematical formula for Nash-Schiff efficiency (NSE). I know Nash-Sutcliffe efficiency as NSE and I’m not sure if it’s the same metrics.

Response: the equations of the used metrics were added in the revised manuscript accordingly. From line 226 to line 229

Round 2

Reviewer 1 Report

Accept